# Status, rivalry and admiration-seeking in narcissism and depression: A behavioral study

Anna Szücs[1,2], Katalin Szanto[1], Jade Adalbert[3], Aidan G. C. Wright[4], Luke Clark[3], Alexandre Y. Dombrovski[1]*

1 Department of Psychiatry, University of Pittsburgh, Pittsburgh, Pennsylvania, United States of America, 2 Department of Psychiatry, University of Geneva, Geneva, Switzerland, 3 Department of Psychology, University of British Columbia, Vancouver, British Columbia, Canada, 4 Department of Psychology, University of Pittsburgh, Pittsburgh, Pennsylvania, United States of America

* dombrovskia@gmail.com

**Data Availability Statement:** All data and analyses have been made publicly available on Zenodo.org. DOI: http://doi.org/10.5281/zenodo.3977825.

## Abstract

Humans seek admiration to boost their social rank and engage in rivalry to protect it when fearing defeat. Traits such as narcissism and affective states such as depression are thought to influence perception of rank and motivation for dominance in opposite ways, but evidence of the underlying behavioral mechanisms is scant. We investigated the effects of dimensionally-assessed narcissism and depression on behavioral responses to social defeat in a rigged video game tournament designed to elicit rivalry (stealing points from opponents) and admiration-seeking (paying for rank). We tested an undergraduate sample (N = 70, mean age = 21.5 years) and a clinical sample of predominantly depressed elderly (N = 85, mean age = 62.6 years). Both rivalry and admiration-seeking increased with time on task and were particularly enhanced in individuals high in narcissism. Participants engaged in more rivalry when pitted against high-ranked opponents, but depression partially mitigated this tendency. Our findings provide behavioral evidence that narcissism manifests in increased rivalry and admiration-seeking during social contests. Depression does not suppress general competitiveness but selectively inhibits upward-focused rivalry.

## Introduction

As primates whose survival and reproduction depend on our standing in a group, we integrate social comparisons, victories, and defeats into an implicit estimate of our social status or rank. Thus, we learn our place and decide how to best improve or maintain it [1–3]. Our implicit rank, i.e. the hierarchical rank manifested through our behavior, is recalibrated following unexpected outcomes of social comparisons against others [4]. Consistently, we tend to avoid confrontations with superiors without a reasonable probability of success [5] and prefer same-level comparisons, which give a better chance of increasing our status while remaining reasonably safe [6]. Individuals, as well as firms, political parties, and sports teams act more competitively when facing similarly-ranked counterparts than stronger ones [7–10].

**Funding:** This work was supported by the National Institute of Mental Health, Maryland, USA (A.D., grant numbers R01MH048463, R01MH100095), (K.Sz., grant number R01MH085651-11).

**Competing interests:** The authors have declared that no competing interests exist.

When threatened with social defeat, people tend to engage in *self-enhancement*, which aims to increase social rank, and in *self-protection*, which aims to avoid further losses by fleeing or fighting back [11]. These competitive behaviors are subject to marked individual differences [12], and can sometimes reach irrational and socially disadvantageous extremes [13,14]. This paper explores behavioral responses to social defeat in narcissism and depressive states, linked by prior self-report and interview studies to opposite patterns of dominant and submissive behaviors [15].

In narcissism, implicit rank is inflated [16] and also closely monitored [17], which may confer certain fitness advantages. Fierce protection of one's dominant status [18] may improve access to resources, and self-inflation improves mating chances [19,20]. These strategies, however, can backfire when affiliation- rather than dominance-driven responses are called for [21], for example during prolonged periods of adversity [22]. The prospect of losing status elicits intense emotions in highly narcissistic individuals [23,24], and leads to counteroffensives to reassert dominance [25]. Back and colleagues termed narcissistic self-enhancement *admiration-seeking* (seeking status through self-promotion), and narcissistic self-protection *rivalry* (antagonizing those perceived as threatening) [26]. Whereas healthy rivalry is attuned to the social rank of the opponent, highly narcissistic persons may engage in indiscriminate rivalry [27], often despite dire moral and financial consequences, and without explicit provocation [28]. At the same time, admiration-seeking takes a comparative form in more narcissistic individuals [29], and is mostly directed towards high-status others [27].

In contrast, depression is thought to shift priorities from achieving dominance to preventing conflict [15,30]. Animals subordinated by aggressive conspecifics become socially avoidant and lack vigor in seeking rewards, a state mediated by plasticity in the mesostriatal pathway and reversed by antidepressants [31,32]. Consistently, subordinate mice display lower anxiety in situations of chronic social defeat than dominant mice [33]. In humans, depressive states appear to deflate one's implicit rank [34], decrease social comparisons and increase submissive behavior when these characteristics are measured by self-reports [35]. Conversely, assertively renegotiating one's role in key relationships is thought to be one mechanism of change in interpersonal therapy for depression [36]. These findings suggest that behavioral and neural plasticity induced by social defeat constitute one component of human depressive states. Thus depressive symptoms are expected to inhibit both rivalry and admiration-seeking in competitive environments.

In summary, while self-report studies describe broadly opposing reactions to social defeat in narcissism and depression, this concept's behavioral manifestations are yet to be tested experimentally. Toward this end, we investigated the effects of dimensionally-measured narcissism and depression on self-protective and self-enhancing behaviors in situations of defeat in a competitive setting. Using a rigged video game tournament paired with a league table, we elicited rivalry (stealing points from opponents), and admiration-seeking (paying to increase one's rank), in an undergraduate sample and a clinical sample of predominantly depressed older adults. To uncover effects of implicit rank (participants' hierarchical status inferred from their behavioral choices), our task manipulated the rank of opponents to examine whether rivalry and admiration-seeking were directed upwards or downwards in the social hierarchy. We expected (H1a) both rivalry and admiration-seeking to increase as a reaction to the cumulative experience of defeat throughout the task; (H1b) narcissism to further enhance both behaviors, prompting a more intense reaction to defeat [26]; and (H1c) depression to dampen them, favoring submissive responses to adversity [35]. Participants' implicit rank was indicated by their level of competitive involvement, i.e. rivalry and admiration-seeking as a function of the opponent's rank. We hypothesized that (H2a) high-ranked opponents would elicit more rivalry and admiration-seeking than low-ranked opponents, given that high-ranked opponents

will be perceived as similar or superior to oneself and thus more threatening [6]. Additionally, (H2b) this effect would be further enhanced by narcissism, which is known to shift implicit rank upward [16], and (H2c) mitigated by depression, which tends to lower implicit rank [34].

## Materials and methods

### Participants

All procedures were approved by the Behavioural Research Ethics Board of the University of British Columbia for Sample 1 and the Institutional Review Board of the University of Pittsburgh for Sample 2. All participants provided their written, informed consent before starting the experiment. Sample 1 included 70 undergraduate students enrolled at the University of British Columbia, Vancouver, Canada (mean age 21.5 years), who participated (in individual sessions) for course credit. Sample 2 included 85 adults aged 50 or older (mean age 62.6 years) participating in the Longitudinal Research Program in Late-life Suicide, in Pittsburgh, United States, a larger ongoing study [37]. Correlates of the present experimental task with suicidal behavior in Sample 2 will be reported elsewhere. Participants from Sample 2 were originally recruited as 25 healthy subjects with no lifetime history of psychiatric disorder and 60 subjects with clinical levels of depression, scoring 14 or higher on the Hamilton Rating Scale for Depression (HRSD) upon study recruitment. The present study treated depression as a continuous variable in both samples. Significant effects were nevertheless tested in a categorical sensitivity analysis in Sample 2 (Fig C in S1 Appendix). Sample characteristics can be found in Table A of S1 Appendix.

### Behavioral task

We used a rigged video game tournament to elicit rivalry and admiration-seeking behaviors under a threat of social defeat (Fig 1; the task is freely available at https://github.com/aszucs/cobra_task_v1).

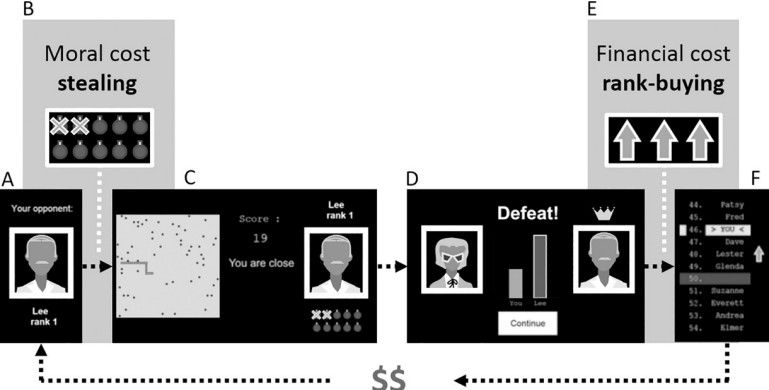

**Fig 1. Task description.** Each of the 24 trials has the following steps: (A), new opponent is displayed with name and league rank (participants are told that these opponents are all previous players whose performance has been prerecorded); (B), 1st outcome measure: willingness to steal points from the opponent's future score in order to increase one's chance of defeating him/her; (C), playing the snake arcade game for 20 seconds, with the goal of gathering as many points (apples) as possible, while one's score (the number of apples caught) is displayed, as well as a message at 10 seconds informing the participant whether he/she is ahead, behind, or the outcome is close; (D), learning the contest's outcome, which defines whether the participant will move 5 ranks up or down in the global ranking. These outcomes are rigged towards a 2:1 defeat to victory ratio occurring in a pseudorandom order; (E), 2nd outcome measure: willingness to buy extra rank; (F), learning current rank in the competition. ($ $), each trial starts with a renewed virtual endowment that the participant can choose to spend on extra ranks. Participants are told that their real money payoff will be computed based on their savings from three random trials. Note. The computer version is in color.

The snake arcade game (adapted from the classic video game to Python 2.7) served as a basis to the competition and was embedded in a tournament interface programmed in Matlab, version 2016b. Each of the 24 trials was divided into a contest phase, when participants played a round of the snake arcade game against different opponents, and a ranking phase, where they gained or lost status in the tournament's league standings. Rivalry was measured by the willingness to steal points from the opponent before the contest (henceforth *point stealing)*. Admiration-seeking was measured by the willingness to pay virtual money to increase one's rank in the league during the ranking phase (henceforth *rank buying)*.

Whereas point stealing had a moral cost of being unfair and unsportsmanlike, participants were told that rank buying had a financial cost: they were instructed that their real money payment would be calculated based on the amount of virtual money saved on each trial, whereas their final standing in the game would not impact the payoff. The end-game payoffs were in fact the same for all participants.

To emulate a real-life social environment, participants were told they were playing against previous study participants whose performance had been recorded. To support this cover story, they chose an avatar to represent themselves in the game and entered a pseudonym; we instructed them not to use their real name, to preserve confidentiality. As a final step, participants were asked to choose whether they wanted to appear in the tournament's league table against future participants. Participants were in fact playing against virtual opponents, arranged in a pseudorandom order and paired with predefined outcomes, having an overall 2:1 defeat to victory ratio.

To mask the rigged outcomes, two additional manipulations were added during the arcade game: the commands from the original snake game were reduced to only the left/right arrow keys, turning the snake 90 degrees left or right on its own axis of reference; second, fewer apples (points) were available for collection on losing trials. The goal of these manipulations was to enhance the game's difficulty, and elicit feelings of frustration and helplessness due to poor performance [38]. Finally, virtual monetary endowments and real money payoffs were adapted to each sample: in Sample 1, participants received 5 CAD (Canadian Dollars) as the trial-by-trial endowment and 7.50 CAD as the overall payoff, while in Sample 2, participants received 20 USD (U. S. Dollars) as the trial-by-trial endowment and 25 USD as the payoff.

The game recorded four subject-dependent variables. The two main outcome measures were participants' choices for point stealing (integers ranging from 1: *no point stealing* to 5: *stealing 10 points from opponent*) and rank buying (integers ranging from 1: *no rank buying* to 5: *buying 5 extra ranks* for half of the trial-by-trial endowment's amount). Additionally, the game recorded participants' own rank (recoded for analysis in an increasing order from 1 = worst rank to 200 = best rank) and scores (number of points on the snake arcade game) on each trial. These two additional subject-dependent variables were used as covariates during the analysis of the main outcome measures (see Statistical Analysis below). Scores were additionally employed in quality checks measuring game performance and task engagement (pages 11–13 in S1 Appendix).

## Other assessments

At the beginning of the task, participants indicated their video game experience (integers ranging from 1: *never played any games including smartphones and tablets* to 5: *playing every day*). After the task, participants answered twelve additional questions about their motivations and impressions of the game that were examined in an exploratory analysis (Fig H in S1 Appendix).

Demographic characteristics (age, sex, ethnicity and household income) were collected at baseline. Due to divergence in data collection, household income was coded as an ordinal

variable in Sample 1 and as logged amounts in Sample 2. Years of education were only assessed in Sample 2, as Sample 1 comprised undergraduates. Ethnicity and household income were missing in three participants of Sample 1.

Narcissism was assessed by the FFNI (Five-Factor Narcissism Inventory) [39] and was treated as a continuous variable, given its dimensional structure corroborated by recent studies [40]. The FFNI's distribution in our two samples can be found in Fig A in S1 Appendix. As a senstivity analysis testing the generalizability of our findings to the more pathological aspects of narcissism [41,42], we additionally used the BPNI (Brief Pathological Narcissism Inventory), a 28-item version of the Pathological Narcissism Inventory [43]. We used total scores in our main analysis, but investigated whether specific dimensions of narcissism were driving the observed behavioral effects in an exploratory analysis including FFNI subscales agentic extraversion, antagonism, and narcissistic neuroticism. The FFNI was missing in four participants of Sample 2. The BPNI was missing in one participant of Sample 1 and two participants of Sample 2.

Depression was assessed by the DASS-21 depression subscale in Sample 1 [44] and the Hamilton Rating Scale for Depression (HRSD) in Sample 2 [45].

Trait dominance was assessed in Sample 2 only by the IPIP-DS (International Personality Item Pool–Dominance Subscale) [46]. We used this measure in an exploratory analysis investigating whether the tendency to thrive for dominance mapped on the behaviors observed with narcissism and depression.

See Table B in S1 Appendix for reliability coefficients of all psychometric measures and Fig B in S1 Appendix for their correlations with task-related variables.

## Procedure

Participants played the task on a laptop computer in Sample 1 and Windows tablets in Sample 2. After they had given written, informed consent to participate, a research assistant walked them through the task instructions, a practice session, and a survey of their prior video game experience, all of which were built in the task. Participants then played the rigged video game tournament for 24 trials. The test administrator stayed in the room but did not watch participants' actions after the first two trials. After finishing the task, participants filled out the additional assessments (DASS-21, FFNI and BPNI scales for Sample 1; FFNI, BPNI and IPIP-DS for Sample 2). In Sample 2, the HRSD was administered by a clinician within a week of the task session in the form of a semi-structured interview.

## Statistical analysis

We examined how (H1a) defeat and (H2a) opponents' rank influenced rivalry and admiration-seeking throughout the task, and how (H1b, H2b) narcissism and (H1c, H2c) depression moderated these relationships.

**Dependent variable and covariates.** All analyses were conducted in R version 3.4. We determined our main analytic approach, dependent variables, variables of interest and covariates at the beginning of data analysis and did not modify them during the subsequent phases of model selection. Point stealing and rank buying were analyzed as continuous, trial-level variables that were person-mean centered around subject means, which resulted in a normal distribution. The person mean-centered scores were then entered as dependent variables in separate linear multi-level models (function *lmer*, package lme4 [47]). As within-subject centering of choices yielded a variance of 0 in the subject intercept, associations between the subject's mean and subject-level variables were not tested. Instead, we examined increases in behavior in response to task manipulations, and interactions with subject-level variables

modulating these relationships. All reported models include age, sex, education (in Sample 2 only), ethnicity, household income, game experience, and depression as co-variates. Unless specified otherwise, main effects' significance did not differ without the inclusion of these variables.

**Model selection.** On each step, model selection was performed using the likelihood-ratio test (function *anova*, package Stats [48]). To diagnose multicollinearity, variance inflation factors adjusted for degrees of freedom were computed and were <2 for all reported effects of retained models.

First, the best-fitting model containing only design variables was selected in both samples. Three experimental condition effects central to our research questions were retained in all models: (H1a) to investigate the effect of defeat, *trial* (time on task) measured overall exposure length to social defeat whereas *most recent outcome* (dummy-coded as 1 for victory and 0 for defeat) measured trial-by-trial positive/negative reinforcement on competitive behavior; (H2a) the level of competitive involvement was measured as increases in behavior in response to the *opponent's rank* (an integer between 1 = worst rank and 200 = best rank). All models additionally included an indicator of performance on the snake arcade game (score on the most recent trial) and participants' current rank. Significant predictors in Sample 1 were retained in models built for Sample 2, even when they were no longer significant in Sample 2. In models predicting rank buying, previous rank-buying choices had to be entered as a covariate, since buying extra rank improved one's own status in the game (see Fig B in S1 Appendix for correlations).

Second, we investigated the effects of narcissism (measured by the FFNI) and depression (measured by the DASS-21 depression subscale in Sample 1 and the HRSD in Sample 2) by adding these psychometric constructs separately to the retained models with design variables, and allowing interactions between them. (H1b, H1c) An interaction effect with *trial* or *outcome* would inform us about the psychometric construct's effect on the behavioral response to defeat, whereas (H2b, H2c) an interaction with *opponent's rank* would suggest an effect on the level of competitive involvement.

Finally, we evaluated the retained models in a pooled analysis, in an effort to verify our results' consistency across age groups and levels of psychopathology. Including *sample* as an independent variable, we ran all selected models described above in a dataset encompassing both samples. The pooled analysis kept all subject-level covariates, which necessitated approximate conversions of household income, education and depression. Household income was recoded as a ranked variable in Sample 2 after conversion of the cut-off values used in Sample 1 from CAD to USD. The DASS-21 depression subscale in Sample 1 and the HRSD in Sample 2 were converted into percentile norms using a software tool developed by Crawford and colleagues in a general population sample [49]. Since percentile norms were not directly available for the HRSD, values of the Carroll Rating Scale for Depression were used instead, which is a self-report version of the HRSD that shares the same items and scoring [50]. Education was assumed to be 13 years in Sample 1.

**Sensitivity analyses.** We conducted four sensitivity analyses:

a. To verify that our findings were truly reflecting behavioral changes arising in response to the task, we assessed the proportion of long-string responders (defined here as participants who repeated the same choice for a given outcome measure throughout the entire task; Table C in S1 Appendix), compared them to the other participants on demographic and psychometric measures (Table D in S1 Appendix) and retested all main findings after excluding participants who engaged in long-string responding on both point stealing and rank buying (Table E in S1 Appendix).

b. Given the well-established role of sex in competitive behaviors [13,51] and the predominance of female participants in both of our samples (resp. 78.6% in Sample 1 and 60.0% in Sample 2; Table A in S1 Appendix), we tested all main findings' robustness to *sex*trial*, *sex*outcome* and *sex*opponent's rank* covariates, added simultaneously to our selected models (Table F in S1 Appendix). We subsequently evaluated all main findings for moderation by sex, one interaction at a time (Table G in S1 Appendix).

c. We tested our main findings in the pooled analysis for sample-level differences by letting sample moderate them.

d. We investigated whether the effects found with the FFNI would generalize to the BPNI by substituting BPNI total scores to FFNI total scores in the models in question.

**Exploratory analyses.** We performed four additional exploratory analyses, in an effort to better situate our main findings within narcissistic dimensions and the task's general dynamics:

a. To investigate which narcissistic dimensions were driving the effects found with the FFNI total score, we tested them with the three FFNI factors agentic extraversion, antagonism and narcissistic neuroticism in the pooled analysis.

b. We explored how our findings of narcissism and depression would compare to the behavioral effects of the tendency to thrive for dominance by testing interactions of trait dominance, as measured by the IPIP-DS in Sample 2, with *trial*, *outcome* and *opponent's rank* (page 10 in S1 Appendix).

c. We analyzed performance (scores on the snake arcade game) in linear mixed-effects models having subject-level intercepts as random effect (pages 11–13 in S1 Appendix). This analysis enabled us to test differences between samples (Fig F in S1 Appendix) and the effects of narcissism and depression on task engagement (Table I and Fig G in S1 Appendix).

d. As a final step in the pooled analysis, we looked at correlations of mean point stealing and rank buying behaviors and psychometric constructs with participants' self-reported motivations and impressions collected at the end of the task (Fig H in S1 Appendix). Our goal was to check whether the observed behaviors and their moderations by psychometric measures was consistent with how participants experienced the task.

## Results

For the reader's convenience, below, we focus on replicated findings illustrated with statistics from the pooled analysis and only describe samplewise models in the main text in the case of inconsistencies. Samplewise findings are further detailed in Figs C and D in S1 Appendix. Table 1 summarizes our main findings.

### (H1) Reaction to defeat

(H1a) We found no effect of the immediately preceding outcome. However, over the task both point stealing and rank buying increased in reaction to defeat, as evidenced by a main effect of *trial* ($\chi^2_1 = 55.33$, $p < .001$ for point stealing; $\chi^2_1 = 20.11$, $p < .001$ for rank buying; Fig 2).

(H1b) Narcissism predicted greater increases in both point stealing and rank buying over time, as indicated by a significant *narcissism*trial* effect ($\chi^2_1 = 7.91$, $p = .005$ for point stealing; $\chi^2_1 = 11.28$, $p < .001$ for rank buying; Fig 3). This effect was not significant in the model

**Table 1. Summary of main findings in the two samples and in the pooled analysis encompassing both.**

| Effects significant in the pooled analysis | Sample 1 (N = 70) | Sample 2 (N = 85) | Pooled (N = 155) |
|---|---|---|---|
| | Coefficient (standard error) | | |
| (i) Reaction to defeat | | | |
| Point stealing | | | |
| Point stealing increases over time | **.133 (.026)**\*\*\* | **.115 (.023)**\*\*\* | **.127 (.017)**\*\*\* |
| Point stealing increases more over time in participants with higher levels of <u>narcissism</u> | .027 (.022) | **.046 (.021)**\* | **.043 (.015)**\*\* |
| Rank buying | | | |
| Rank buying increases over time | **.107 (.025)**\*\*\* | **.045 (.023)**\* | **.075 (.017)**\*\*\* |
| Rank buying increases more over time in participants with higher levels of <u>narcissism</u> | **.057 (.024)**\* | **.043 (.021)**\* | **.051 (.015)**\*\*\* |
| (ii) Level of social comparisons | | | |
| Point stealing | | | |
| Point stealing increases against high-ranked opponents | **.068 (.024)**\*\* | **.095 (.022)**\*\*\* | **.085 (.016)**\*\*\* |
| Point stealing increases more against high-ranked opponents after having performed well on the arcade game | **.046 (.021)**\* | **.051 (.019)**\*\* | **.032 (.014)**\* |
| Point stealing does **not** increase against high-ranked opponents in highly <u>depressed participants</u> | -.031 (.024) | **-.064 (.019)**\*\*\* | **-.057 (.014)**\*\*\* |
| Rank buying | | | |
| Rank buying increases more over time against high-ranked opponents | **.059 (.021)**\*\* | .008 (.020) | **.030 (.014)**\* |

Significant effects are in bold.

\*, p < .05

\*\*, p < .01

\*\*\*, p < .001.

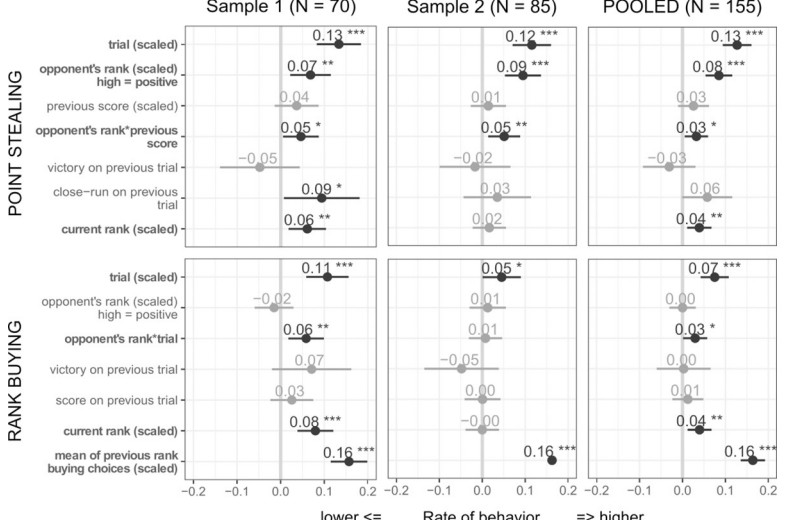

**Fig 2. Models with design variables predicting point stealing and rank buying in the two samples and the pooled analysis.** Estimates of demographic covariates present in the models are not displayed (age, sex, ethnicity, household income, education, game experience, depression); effects significant in the pooled analysis are in bold, significant coefficients within each table appear in darker gray. The significant positive effect of *trial* (i.e. time on task) on both point stealing and rank buying behaviors can be interpreted as a reaction to the increasing exposure to defeat (given the rigged outcomes). *Opponent's rank* and player's *previous score*\**opponent's rank* were consistent predictors of point stealing across samples, indicating a preference for upward directed rivalry, especially after having performed well on the snake arcade game. The mean of previous rank-buying choices was included as a covariate in the model predicting rank buying since buying extra rank inflated participant's own rank (see Methods - Statistical Analysis–Model Selection). Points and numbers indicate estimates of fixed effects; horizontal bars represent standard errors. Legend: \*, p < .05; \*\*, p < .01; \*\*\*, p < .001.

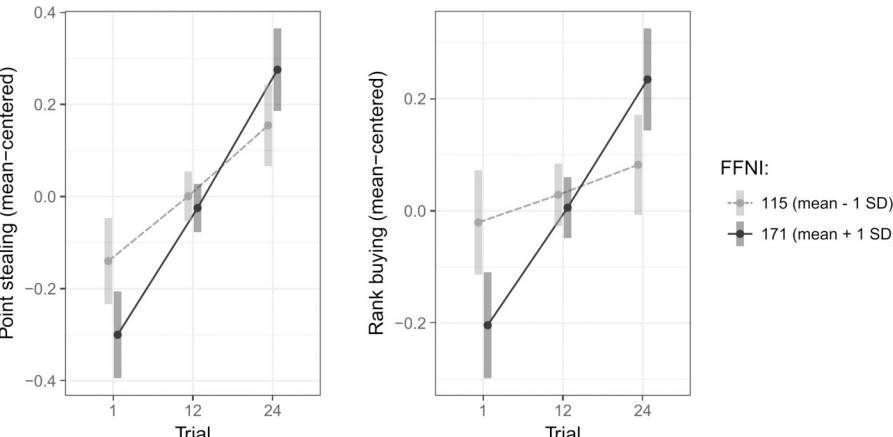

**Fig 3.** Significant narcissism*trial interactions predicting point stealing (left) and rank buying (right) indicating that more narcissistic individuals tended to increase both behaviors in response to the cumulative experience of defeat. The above effects were robust to subject-level covariates (age, sex, education, ethnicity, household income, game experience and depression). Points are estimates from the corresponding regression model at the indicated values; vertical bars represent 95% confidence intervals. Legend: FFNI, Five-Factor Narcissism Inventory.

predicting point stealing in Sample 1, but shared a similar pattern in all other cases (Fig C, Panel A in S1 Appendix).

(H1c) Depression did not influence point stealing and rank buying over time.

## (H2) Level of competitive involvement

(H2a) With respect to point stealing, our models with design variables indicated that people stole more points when pitted against high-ranked opponents ($\chi^2_1$ = 28.33, p < .001) and even more so when facing a high-ranked opponent after achieving a high score on the previous round (opponent's rank*previous score: $\chi^2_1$ = 5.42, p = .020). Rank buying was not higher against high-ranked opponents overall, but did increase more against them over time, as evidenced by an *opponent's rank*trial* interaction ($\chi^2_1$ = 4.22, $p$ = .040). This effect was not significant in Sample 2.(H2b) No moderation effect was present between opponent's rank and narcissism in the pooled analysis. Initial *narcissism*opponent's rank*trial* and *narcissism*opponent's rank* effects predicting rank buying were only found in Sample 1 (resp. $\chi^2_1$ = 4.40, $p$ = .036 and $\chi^2_1$ = 4.75, $p$ = .029; Fig D in S1 Appendix) and were therefore not retained among our main findings.(H2c) A *depression*opponent's rank* interaction predicting point stealing ($\chi^2_1$ = 16.79, $p$ < .001; Fig 4) evidenced a consistent loss of sensitivity to opponents' rank among more depressed participants. After the inclusion of subject-level covariates, this effect fell short of significance when tested separately in Sample 1 (Fig C, Panel B in S1 Appendix), where depression scores were tightly distributed around the population average (Table A in S1 Appendix). In Sample 2, the effect was robust to covariates and remained present when depression was analyzed categorically ($\chi^2_1$ = 9.33, $p$ = .002; Fig C, Panel C in S1 Appendix).

## Sensitivity analyses

(a) The proportion of long-string responders was consistent across samples. Long-string responders on both outcome measures represented respectively 12 and 13% of participants in Sample 1 and 2 (Table C in S1 Appendix). They did not differ from other subjects, with the exception of none being African-American in Sample 2 vs. 18.7% in the rest of the sample

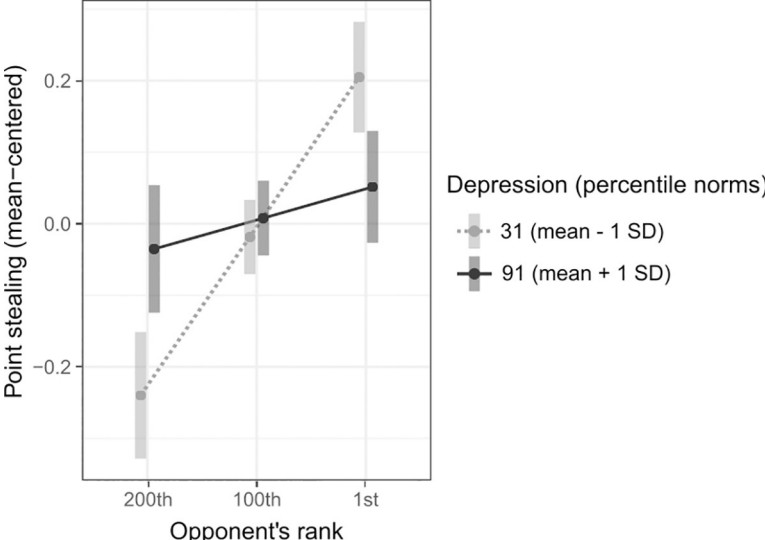

**Fig 4. Significant depression*opponent's rank interaction predicting point stealing.** By contrast to participants high on trait dominance, more depressed individuals failed to adjust point stealing to their opponents' rank. Points are estimates from the corresponding regression model at the indicated values; vertical bars represent 95% confidence intervals.

(Table D in S1 Appendix). Excluding long-string responders from the analysis did not change any of our main findings (Table E in S1 Appendix).

(b) Including *sex*trial*, *sex*outcome* and *sex*opponent's rank* did not influence our main findings either in the pooled analysis, or in the individual samples, with the exception of the main effect of trial on rank buying that lost significance in Sample 2 but maintained comparable effect magnitude to the principal model (Table F in S1 Appendix). Interaction effects with sex were not significant in models predicting point stealing (Table G in S1 Appendix). With respect to rank buying, negative *sex*trial* and *sex*trial*opponent's rank* effects emerged in the pooled analysis (resp. $\chi^2_1 = 6.72$, p = .010 and $\chi^2_1 = 7.96$, p = .0047), the former being only significant in Sample 1 ($\chi^2_1 = 7.22$, p = .007) and the latter in Sample 2 ($\chi^2_1 = 4.71$, p = .030). Sex did not moderate the effects of narcissism or depression.

(c) Our main findings did not generally differ across samples in the pooled analysis, with the following exceptions: a *sample*opponent's rank* interaction predicting point stealing ($\chi^2_1 = 4.70$, p = .030) and a *sample*trial* interaction predicting rank buying ($\chi^2_1 = 5.10$, p = .024) indicated similar effect directions in both samples, but a greater effect magnitude, respectively, in Sample 2 for opponent's rank predicting point stealing and in Sample 1 for trial predicting rank buying.

(d) Similar *narcissism*trial* effects were found with the BPNI as with the FFNI in the pooled analysis ($\chi^2_1 = 7.65$, $p = .006$ for point stealing; $\chi^2_1 = 4.33$, $p = .038$ for rank buying; Table 2). In the individual samples, this effect did only reach significance for point stealing in Sample 1 ($\chi^2_1 = 3.97$, $p = .046$).

## Exploratory analyses

(a) With respect to FFNI subscales, agentic extraversion was associated with both point stealing and rank buying (resp. $\chi^2_1 = 7.28$, $p = .007$ and $\chi^2_1 = 4.28$, $p = .039$), antagonism with rank buying ($\chi^2_1 = 5.54$, $p = .019$) and narcissistic neuroticism with none of the behaviors.

(b) In Sample 2, where the IPIP-DS was administered, trait dominance behaved similarly to narcissism and opposedly to depression (Table H and Fig E in S1 Appendix): it increased

**Table 2.  Effects of BPNI narcissism and of lower-level FFNI dimensions in the pooled analysis.**

| Main effects of interest for narcissism, as measured by the FFNI | FFNI | BPNI |
|---|---|---|
| | Coefficient (standard error) | |
| Reaction to defeat (narcissism*trial interaction) | | |
| Point stealing | | |
| TOTAL SCORE | .043 (.015)** | .043 (.016)** |
| Agentic extraversion | .041 (.015)** | - |
| Antagonism | .024 (.015) | |
| Narcissistic neuroticism | .018 (.015) | |
| Rank buying | | |
| TOTAL SCORE | .063 (.017)*** | .032 (.016)* |
| Agentic extraversion | .031 (.015)* | - |
| Antagonism | .035 (.015)* | |
| Narcissistic neuroticism | .015 (.015) | |

FFNI, Five-Factor Narcissism Inventory; BPNI, Brief Pathological Narcissism Inventory

*, p < .05

**, p < .01

***, p < .001.

point stealing and rank buying over time, as indicated by a *trait dominance*trial* interaction
($\chi^2_1 = 7.45$, $p = .006$ for point stealing; $\chi^2_1 = 5.90$, $p = .015$ for rank buying), and increased the
tendency to engage in point stealing against high-ranked opponents, as evidenced by a positive
*trait dominance*opponent's rank* interaction predicting point stealing ($\chi^2_1 = 10.90$, $p = .001$).

(c) Task performance, as measured by scores on the snake arcade game increased with time
($\chi^2_1 = 104.95$, $p < .001$), albeit less steeply in Sample 2 (*trial*sample* interaction in the pooled
analysis: $\chi^2_1 = 12.44$, $p < .001$). Narcissism further accentuated performance over time ($\chi^2_1 =
7.65$, $p = .006$). This effect was driven by agentic extraversion ($\chi^2_1 = 8.98$, $p = .003$) and to a lesser
extent by antagonism ($\chi^2_1 = 4.12$, $p = .042$). Antagonism however also predicted lower scores
overall ($\chi^2_1 = 4.37$, $p = .037$). Depression predicted lower scores overall ($\chi^2_1 = 4.53$, $p = .033$),
without influencing improvement over time. For further details, see pages 11–13 in S1 Appendix.

(d) Correlations of mean behaviors and psychometric measures with participants' self-
reported feedback can be found in Fig H in S1 Appendix. Mean point stealing and rank buying
respectively correlated at .24 and .26 with striving for status (question M5) and at .31 and .20
with striving for victory (question M8). Seeking to outperform others (question M3) and avoid-
ing being worse than everyone else (question M4) additionally correlated with mean point steal-
ing at respectively .25 and .17. Judging one's own performance favorably correlated with rank
buying at .20 (question A4). It also had a positive correlation with agentic extraversion at .23
and a negative one with narcissistic neuroticism at -.16. Depression negatively correlated with
enjoyment of the task at -.26 (question A3) and with believed fairness of opponents at -.23
(question A2), whereas agentic extraversion positively correlated with striving for status at .27
(question M5), striving for victory at .20 (question M8), seeking to outperform others at .24
(question M3) and seeking revenge at .17 (question M6). Antagonism correlated with seeking
revenge at .26 (question M6) and striving for victory at .16 (question M8).

## Discussion

We used a rigged video game tournament experiment to elicit rivalry and admiration-seeking
behaviors and investigate the level of social comparisons under the threat of defeat. We

observed an increase of both behaviors with time on task, which was further enhanced by narcissism. In contrast, depression did not inhibit these behaviors, against our prediction. With respect to the level of competitive engagement, we observed no consistent effect of narcissism but found that upward-focused rivalry was inhibited by depression. Taken together, our findings are consistent with the maintainance of social status constituting a general human motivation [52], and further suggest that this goal is moderated by narcissism and depression on two distinct levels: while narcissism primarily increases the intensity of rivalry and admiration-seeking, depression influences rivalry's objective, by inhibiting upward-focused ambitions.

Experimentally corroborating the rivalry/admiration theory [26], more narcissistic persons were more intensely motivated to protect and promote their implicit rank in the face of social defeat (Fig 3). These behaviors scaled in similar ways with trait dominance as with narcissism (Fig E in S1 Appendix), which was consistent with overlapping dominance-driven motivations between the two constructs, as outlined in the Dominance Behavioral System [15]. Narcissism also correlated with faster improvement in performance on the snake arcade game (Fig G in S1 Appendix), corroborating the higher competitiveness and task engagement found in narcissism [53], especially on tasks where good performance provides the opportunity to self-enhance [54].

The fact that participants high in narcissism started the task with lower rates of rivalry and admiration-seeking than their less narcissistic counterparts (Fig 3) is consistent with these behaviors' compensatory role. Individuals perceiving themselves as powerful have been found to resort to aggression primarily when feeling incompetent and threatened in their self-view [55], and since narcissism has been linked to overestimating one's future performance [56], it is likely that more narcissistic participants only resorted to alternative pathways once their own performance appeared insufficient to achieve dominance.

Consistent with Back and colleagues' theory [26], we observed that rivalry and admiration-seeking behaviors were mostly driven by the grandiose dimensions of narcissism (the FFNI dimensions agentic extraversion, and to a lesser extent antagonism; Table 2). Whereas these behaviors also occured in more pathological forms of narcissism, as measured by the BPNI, they had no association with narcissistic neuroticism on the FFNI.

In Back's conceptualization, admiration-seeking roughly corresponds to agentic extraversion and rivalry to antagonism [25]. However, in our study, antagonism only enhanced rank buying, our behavioral measure of admiration-seeking (Table 2). Rivalry in our paradigm did not, in fact, include components of reactive anger, a core facet of FFNI antagonism [39], since point stealing occurred *before* playing against a given opponent. Further, based on the instructions, opponents were presented as *previous* participants and therefore were not handicapped by point stealing in real time. On the other hand, rank buying took place right *after* learning the trial's outcome and thus likely acquired a reactive component. This is also corroborated by the increase of rank buying against high-ranked opponents over time (Table 1). It nevertheless remains unclear whether a similar pattern would occur with self-reported rivalry, since FFNI antagonism and rivalry measured by the Narcissistic Admiration and Rivarly Questionnaire (NARQ) are not fully overlapping constructs [25,26]. Rivalry in our paradigm matches Back's definition, namely a willingness to surpass and devalue others in a socially insensitive way [25], and is consistently correlated with the self-reported motivation of outperforming others (Fig H in S1 Appendix). However, revenge-orientation has a strong association with rivalry in prior research [26], and it is likely that our behavioral measure of rivalry does not capture the constructs' more antagonistic aspects.

Consistently with our second hypothesis with respect to the level of competitive involvement, rivalry was preferentially upward-focused (Fig 2). Although our paradigm did not test participants' preferred level of social challenges, which may have reflected their implicit rank

more directly [6], a strong performance on the previous trial further accentuated rivalry against high-ranked opponents, supporting an upward shift in implicit rank and an increase in assertiveness after positive prediction errors about one's capability [3,4].

In contrast, depression inhibited upward-directed rivalry (Fig 4). This was not explained by a lack of engagement in the task, as indicated by intact performance improvement in more depressed individuals (Table I in S1 Appendix). Nor was it due to decreased competitiveness, since depression did not inhibit point stealing. Depressed individuals exhibited rivalry in a manner insensitive to others' rank and were not selectively motivated to dominate high-status others, conversely to participants high in trait dominance, whose competitiveness primarily manifested against high-ranked opponents. This is consistent with prior research, finding depression to correlate positively with performance-avoidance goals, i.e. trying not to under-perform compared to others, and negatively with performance-approach goals, i.e. aiming to outperform others [57]. Further, depressive individuals' insecurity about their own social rank has been found to prompt competitiveness primarily out of fear of inferiority and of subse-quent rejection [58], contrasting with individuals perceiving themselves as powerful, who tend to pay little attention to low-power others [59]. Thus, the observed behavioral patterns align with the social competition hypothesis of depression, stating that by down-regulating domi-nance motivations, depression grants survival to presumably weaker individuals in a hostile environment [30].

The strengths of our study include the nuanced and novel experimental assessment of com-petitive behavior dynamics and sensitivity to social hierarchy. Our findings' consistency across two very different samples in terms of age and psychopathology and their robustness to sex dif-ferences and other demographic covariates add to their generalizability. As limitations, we note the moderate sizes of each individual sample and the absence of a trait dominance mea-sure in Sample 1. In addition, the weaker manipulation effects in Sample 2 could be due to older adults' lower cognitive functioning and/or relative inexperience with the video game interface. The competition was limited to the duration of the task (participants did not have access to the league table once they finished playing) and did not take place in real time (oppo-nents were said to be past players), which may have taken away some of participants' motiva-tion to perform well. It is further possible that some participants did not believe in the deceptive elements incorporated in the task and therefore experienced less affective involve-ment in the competition. Our participants were not specifically sampled for high/pathological levels of narcissism, since most of the conceptual frameworks relevant to our paradigm have focused on normally distributed subclinical narcissistic traits [3,26,60]. Future research should nonetheless explore how our behavioral findings map on narcissism's most pathological forms, namely narcissistic personality disorder. Finally, although the admiration and rivalry framework developed by Back and colleagues heavily informed our thinking when building our paradigm [26], the NARQ was not used in the current study. This inventory should be included in future works to calibrate our behavioral measures of rivalry and admiration to self-report items.

Our approach captured behaviors that map on narcissistic and depressive semiology, such as narcissistic dominance-striving or depressive dominance-inhibition, and dissected how these tendencies are expressed in social defeat. The decisional biases highlighted by our find-ings are likely reflected in everyday behavior in both general and clinical populations. The fact that social defeat and status can be successfully manipulated experimentally trial-by-trial in real time opens the way for the study of their physiological and neural correlates, going beyond the classic bargaining games predominantly employed to date.

## Supporting information

**S1 Appendix. All supplemental figures and tables.**
(PDF)

## Acknowledgments

The authors acknowledge Laura Kenneally for help with the manuscript as well as Maria Alessi, Amanda Collier, Laura Kenneally and Michelle Perry for their work on data collection.

## Author Contributions

**Conceptualization:** Anna Szücs, Katalin Szanto, Aidan G. C. Wright, Luke Clark, Alexandre Y. Dombrovski.

**Data curation:** Anna Szücs.

**Formal analysis:** Anna Szücs, Alexandre Y. Dombrovski.

**Funding acquisition:** Katalin Szanto, Alexandre Y. Dombrovski.

**Investigation:** Anna Szücs, Jade Adalbert.

**Methodology:** Anna Szücs, Katalin Szanto, Aidan G. C. Wright, Luke Clark, Alexandre Y. Dombrovski.

**Resources:** Katalin Szanto, Jade Adalbert, Luke Clark, Alexandre Y. Dombrovski.

**Software:** Anna Szücs.

**Supervision:** Katalin Szanto, Alexandre Y. Dombrovski.

**Visualization:** Anna Szücs.

**Writing – original draft:** Anna Szücs.

**Writing – review & editing:** Katalin Szanto, Aidan G. C. Wright, Luke Clark, Alexandre Y. Dombrovski.

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
