## [Decision Letter · Decision Letter 0]

5 Oct 2020

PONE-D-20-25628

Status, rivalry and admiration-seeking in narcissism and depression: a behavioral study

PLOS ONE

Dear Dr. Dombrovski,

Thank you for submitting your manuscript to PLOS ONE. After careful consideration, we feel that it has merit but does not fully meet PLOS ONE’s publication criteria as it currently stands. Therefore, we invite you to submit a revised version of the manuscript that addresses the points raised during the review process. Both reviewers were positive on the manuscript each providing enumerating lists of changes that I will not belabor here. I would, for instance, like to see more streamlining/efficiency as noted by Reviewer 2; I prefer less philosopshizing and more science in research along with not trying to oversell/step the data. Conservative approaches to conclusions and analyses are preferable given the state of modern social psychology.

We look forward to receiving your revised manuscript.

Kind regards,

Peter Karl Jonason

Academic Editor

PLOS ONE

Journal Requirements:

Reviewers' comments:

Reviewer's Responses to Questions

**Comments to the Author**

1. Is the manuscript technically sound, and do the data support the conclusions?

Reviewer #1: Yes

Reviewer #2: Yes

2. Has the statistical analysis been performed appropriately and rigorously? 

Reviewer #1: I Don't Know

Reviewer #2: Yes

3. Have the authors made all data underlying the findings in their manuscript fully available?

Reviewer #1: Yes

Reviewer #2: Yes

4. Is the manuscript presented in an intelligible fashion and written in standard English?

Reviewer #1: Yes

Reviewer #2: Yes

5. Review Comments to the Author

Reviewer #1: This is Nick Holtzman at Georgia Southern University. I sign all my reviews. I learned a lot from this paper, and it is clear that the authors are dedicated scholars with technical skills. I do have some critiques, concerns, questions, and edits; I hope these comments prove useful and constructive. The comments are roughly in descending order of importance, and I provide a general recommendation at the end.

1. The main concern is about pre-registration. Although the sensitivity analyses and the replication across samples appears convincing of the general pattern, it is conceivable that the pattern could have been teased out through taking the garden of forking paths (as Andrew Gelman calls it). Because pre-registration is no longer possible for this study, it seems that the only way to convince readers that the garden of forking paths was not traveled is to explicitly state which analyses were conducted. If the authors only conducted the present set of analyses, then please indicate that. If other analyses were conducted but were not presented, then please indicate which ones were conducted. Because this paper was not pre-registered—and especially given the small sample sizes—it is necessary to allay any concerns about p-hacking. This is further complicated by the large number of covariates in the model (see lines 223-224). It’s not that covariates are a problem by themselves; it’s that the possibility of p-hacking and alternative covariates is especially problematic in small samples.

2. One oddity in the results is that, in the undergraduate sample, the association between depression and narcissism was positive. This usually doesn’t happen in younger samples (e.g., see the fascinating SPPS paper by Patrick Hill and Brent Roberts). I attribute this to statistical error, but it may have constrained the ability to tease out fully clear differential results for narcissism and depression. (You can imagine an extreme case where narcissism and depression are correlated .80, and thus it would be nearly impossible to get differential results for the two constructs).

3. From a measurement standpoint, I was confused about the chosen measures and why Admiration and Rivalry weren’t assessed directly. I bet that if Mitja Back read this, he would say the same thing. Maybe one of the measures the authors did use could be converted to admiration and rivalry—I’m not sure. This would clearly require a lot of additional analyses, but it would make the line of reasoning straightforward.

4. The phrase “implicit rank” carries a measurement connotation of the implicit association test, which has seen better days. Is this necessarily implicit? If the idea is about self-perceived rank, then that phrase could be used instead.

5. Please unpack the terse verbiage in H2a and H2b; two things would help me understand this more readily: first, eliminating the dashes between words like upward-focused (which needs to be explained), and second, providing an example to make it more concrete.

6. In the participants section, use “included” instead of “was composed of”, the latter of which involves passive voice.

7. On line 136-137, specify what type of payment. There is an imaginary currency and a real currency at play, so please be more specific in this sentence.

8. Line 151: Spelling error on difficulty.

9. I was confused about rankings, specifically in line 162. Usually, being #1 is best, but here, being #200 is best, right? Also the wording is confusing to me, because “highest” implies best or most superior, but usually that is a word that belongs to the first-ranked individual. One way to simplify this and retain the numbers used in analytics, would simply be to say “best” and “worst”.

10. Please cite the authors of the statistics packages in R (e.g., on line 228).

11. I am not sure what line 240-241 means where the authors write that “significant predictors … were maintained … even when non-significant.” Please clarify.

12. The phrase “stereotypical response rates” is new to me. Is this a common phrase in the literature that I’ve missed? I am accustomed to seeing a phrase like “long-string analysis” (Curran, 2016, JESP). Either an explanation of the phrase or switching the phrase would be welcomed.

13. On line 509, I’m not sure what the dash means after dominance.

14. Figure 2 must have taken a long time to create, and it looks excellent. Nice work.

15. Figures 3 & 4 could be improved slightly by making sure the bars do not overlap. There is a setting for this in R so that you can stagger and space the bars.

16. There are a couple of papers that are pertinent that could be cited to round out the literature review:

a. Wallace and Baumiester had a paper on perceived opportunity for glory in narcissists.

b. Fast and Chen had a paper: https://doi.org/10.1111/j.1467-9280.2009.02452.x

17. In general, I would recommend pulling Chen’s papers from the literature to see if there are any other hints that would be helpful. Her work is highly relevant here.

All told, my main recommendation is for the authors to consider whether their results will hold up in the long run. This will require some reflection on the analytic path that they authors took. I think it is necessary to be explicit about whether any other analytic paths were taken, and if the number of paths is numerous, then it is probably best to hedge on the conclusiveness of this project. That being said, to the editor, I would recommend soliciting a response from the authors involving a reflection on pre-registration, an honest self-assessment of analytic paths taken, and whether the authors think this set of results will hold.

Reviewer #2: Title: “Status, rivalry and admiration-seeking in narcissism and depression: A behavioral study”

I was excited to review this manuscript because it concerned an extremely interesting research question. I was impressed that the authors used an interesting procedure to capture the dynamics surrounding status while engaging in a competitive video game task. I think the manuscript has the potential to make a small but interesting contribution to the literature.

Below are my specific suggestions and concerns regarding the manuscript:

1. My broad reaction is that the authors may be trying to do so much with this manuscript that it may be difficult for readers to extract the most meaningful information. As a result, my advice for the authors would be for them to streamline the manuscript so that it is more focused. As it currently stands, the manuscript is a bit messy and confusing because the authors have so much happening in the manuscript that it is hard to follow. For example, the authors included analyses concerning trait dominance even though the Introduction does not really provide a strong rationale for doing so since it was focused largely on narcissism and depression. Further, the authors only collected a measure of trait dominance in Sample 2 but not Sample 1 which suggests that they did not anticipate trait dominance being a central feature of this work. The authors should either drop trait dominance from their analyses (and maybe include a footnote regarding the analyses concerning trait dominance for Sample 2) or revise the Introduction so that it gives a bit more attention to trait dominance.

2. I think the results concerning narcissism were the most interesting in the manuscript but it is hard to follow everything because there are so many different conceptualizations of narcissism included in the manuscript. My advice would be to simplify things. For the FFNI, it probably makes the most sense to focus on the three-dimensional model (i.e., extraversion, antagonism, and neuroticism). If the authors think it is important to also report the results for the total FFNI score and the grandiose and vulnerable dimensions, then it may be better to do that in a footnote.

3. It may be helpful for the authors to provide a bit more information concerning the rationale for their hypotheses. I think the authors have very interesting ideas but it may be helpful for readers if they provide a little more information to clarify their logic for some of the predictions.

4. The Results section was difficult to follow. I think the authors could make it far more readable by streamlining the number of variables they are including in their analyses so I hope they consider that approach.

5. I was a bit confused by the operationalization of “social comparison” in the manuscript. If I am understanding it correctly, the authors used increases in point-stealing and rank-buying in conjunction with the rank of the opponent to capture “social comparison.” I think the construct that is being captured by the authors is interesting but I am not quite sure that it is really social comparison.

6. The rigged video game tournament is certainly an interesting approach for capturing these sorts of dynamics. I applaud the authors for their efforts to use this sort of approach. However, I think that some of the limitations of this approach deserve a bit more attention in the Discussion. The fact that point-stealing took place before playing an opponent whereas rank-buying took place after playing an opponent is an issue. The authors acknowledge that issue briefly but I think it deserves more attention. Also, the issue that participants would never see the leader board again after their session makes it a bit odd and may be a somewhat weak situation with regard to motivating individuals to consider point-stealing or rank-buying.

7. The pattern of results for FFNI antagonism were surprising. The authors briefly address this issue in the Discussion but it may warrant a bit more attention and consideration from the authors.

6. PLOS authors have the option to publish the peer review history of their article (what does this mean?). If published, this will include your full peer review and any attached files.

Reviewer #1: No

Reviewer #2: No

---

## [Author Response · Author response to Decision Letter 0]

19 Nov 2020

# Editorial comments

E-C1. Please ensure that your manuscript meets PLOS ONE's style requirements, including those for file naming. The PLOS ONE style templates can be found at

A: We have formatted the manuscript according to the guidelines.

E-C2. Please include captions for your Supporting Information files at the end of your manuscript, and update any in-text citations to match accordingly. Please see our Supporting Information guidelines for more information: http://journals.plos.org/plosone/s/supporting-information.

A: We have included the caption for our Supporting Information file (S1 Appendix) at the end of the manuscript and formatted all references to supplemental Tables/Figures according to the guidelines.

# Reviewer 1

R1-C1. The main concern is about pre-registration. Although the sensitivity analyses and the replication across samples appears convincing of the general pattern, it is conceivable that the pattern could have been teased out through taking the garden of forking paths (as Andrew Gelman calls it). Because pre-registration is no longer possible for this study, it seems that the only way to convince readers that the garden of forking paths was not traveled is to explicitly state which analyses were conducted. If the authors only conducted the present set of analyses, then please indicate that. If other analyses were conducted but were not presented, then please indicate which ones were conducted. Because this paper was not preregistered—and especially given the small sample sizes—it is necessary to allay any concerns about p-hacking. This is further complicated by the large number of covariates in the model (see lines 223-224). It’s not that covariates are a problem by themselves; it’s that the possibility of p-hacking and alternative covariates is especially problematic in small samples.

A: We agree that preregistration would have been desirable for this study employing a new behavioral experiment. To be fully transparent, the paradigm was developed in the context of a NIMH grant supplement (R01MH085651S1) exploring how frustrated dominance experimentally instantiated, as threat of social defeat can lead to detrimental decisions in people with specific vulnerability factors, such as narcissistic personality traits. The wider aim of the project is to investigate a potential pathway to late-life suicidal behavior in older adults. The present manuscript constitutes the first report validating the experimental paradigm, hence the inclusion of the undergraduate sample.

If we understand the reviewer correctly, the concern is that the predictors and models reported here could have been cherry-picked from a larger set. From the start, the study aimed to test manipulation effects and identify effects of narcissism and depression, as an initial step towards understanding the behavioral mechanism underlying their contribution to the suicidal crisis. We have not investigated other psychological constructs besides narcissism, depression and trait dominance; we will report on behavioral correlates of late-life suicidal behavior in a second manuscript, once data collection in Pittsburgh reaches an N of 170 (twice the current N of 85), a sample size estimated sufficient to contrast suicidal and non-suicidal depressed participants.

It is true that although frustrated dominance was central to our framework, we did not have a specific a priori hypothesis for trait dominance per se, and we did not assess trait dominance in the Vancouver undergraduate sample for this reason. To focus more on a priori hypotheses, we have now moved the effects of trait dominance from our main findings to the Exploratory analyses section (see also our answer to Reviewer 2’s Comment 1).

With respect to the analytic strategy, our models’ structure and the covariates we used were defined from the start, based on our hypotheses. Regarding the covariates, first, we would like to emphasize that the effects of interest did not change when the covariates were omitted. Second, we used a similar set of covariates (with the exception of gaming experience, which is uniquely relevant to the current experiment) to statistically control for confounds as in our other recent papers (1-4). We hope that these references can reassure the reviewer that this is our standard way of dealing with potential confounds.

Following the reviewer’s suggestion, we have added a statement in the Methods indicating that: “We determined our main analytic approach, dependent variables, variables of interest and covariates at the beginning of data analysis and did not modify them during the subsequent phases of model selection.” (lines 227-229)

(1) Dombrovski, A. Y., Aslinger, E., Wright, A. G. C., & Szanto, K. (2018). Losing the battle: Perceived status loss and contemplated or attempted suicide in older adults. International Journal of Geriatric Psychiatry, 33(7), 907–914. https://doi.org/10.1002/gps.4869

(2) Kenneally, L. B., Szücs, A., Szántó, K., & Dombrovski, A. Y. (2019). Familial and social transmission of suicidal behavior in older adults. Journal of Affective Disorders, 245, 589–596. https://doi.org/10.1016/j.jad.2018.11.019

(3) Szücs, A., Szanto, K., Wright, A. G. C., & Dombrovski, A. Y. (2020). Personality of late‐ and early‐onset elderly suicide attempters. International Journal of Geriatric Psychiatry, 35(4), 384–395. https://doi.org/10.1002/gps.5254

(4) Szanto, K., Galfalvy, H., Kenneally, L., Almasi, R., & Dombrovski, A. Y. (2020). Predictors of serious suicidal behavior in late-life depression. European Neuropsychopharmacology. https://doi.org/10.1016/j.euroneuro.2020.06.005

R1-C2. One oddity in the results is that, in the undergraduate sample, the association between depression and narcissism was positive. This usually doesn’t happen in younger samples (e.g., see the fascinating SPPS paper by Patrick Hill and Brent Roberts). I attribute this to statistical error, but it may have constrained the ability to tease out fully clear differential results for narcissism and depression. (You can imagine an extreme case where narcissism and depression are correlated .80, and thus it would be nearly impossible to get differential results for the two constructs).

A: Narcissism was assessed by the Five-Factor Narcissism Inventory (FFNI) in our samples, which is a measure encompassing both grandiose and vulnerable narcissistic dimensions.

However, in their article entitled Narcissism, Well-Being, and Observer-Rated Personality Across the Lifespan (https://doi.org/10.1177/1948550611415867), Hill and Roberts use the 40-item version of the Narcissistic Personality Inventory (NPI), which only assesses grandiose narcissism, and even then, much of its content (e.g., Leadership) is notorious for significantly correlating with adaptive functioning. Indeed, much has been written about the challenges of using the NPI given its heterogeneous content (5). We are confident that this explains the discrepancy between their findings and ours in young adults. Grandiose narcissism in our undergraduate sample has a close to zero correlation with depression (see figure in the Response to reviewers.pdf file).

(5) Ackerman, R. A., Witt, E. A., Donnellan, M. B., Trzesniewski, K. H., Robins, R. W., & Kashy, D. A. (2011). What does the narcissistic personality inventory really measure? Assessment, 18(1), 67-87.

R1-C3. From a measurement standpoint, I was confused about the chosen measures and why Admiration and Rivalry weren’t assessed directly. I bet that if Mitja Back read this, he would say the same thing. Maybe one of the measures the authors did use could be converted to admiration and rivalry—I’m not sure. This would clearly require a lot of additional analyses, but it would make the line of reasoning straightforward.

A: Thank you for raising this point. As mentioned in response to the previous comment, our original objective was to relate our behavioral measures to the global construct of narcissism, not specifically to its grandiose dimension. When designing our study protocol, we therefore chose two well-established measures that encompassed both narcissistic dimensions: the Five-Factor Narcissism Inventory (FFNI) and, as a sensitivity measure, the Brief Pathological Narcissistic Inventory (BPNI). However, since the behaviors we capture are most specifically formalized in Mitja Back’s Admiration and Rivalry framework, we nevertheless agree with the reviewer that it is an important direction for future research to test the task with the Narcissistic Admiration and Rivalry Questionnaire (NARQ).

We know of no validated method to derive these dimensions from the FFNI or the BPNI. However, since Mitja Back conceptualizes Admiration as approximately similar to FFNI agentic extraversion and Rivalry to FFNI antagonism,6 we have extended the part of the discussion where we elaborate on our findings about these lower-level FFNI dimensions and their possible links to Rivalry and Admiration (please, see our answer to Reviewer 2’s Comment 7 and lines 492-511 in the manuscript for more detail) (6).

(6) Back, M. D. (2018). The Narcissistic Admiration and Rivalry Concept. In A. D. Hermann, A. B. Brunell, & J. D. Foster (Eds.), Handbook of Trait Narcissism (pp. 57–67). Springer International Publishing. https://doi.org/10.1007/978-3-319-92171-6_6

R1-C4. The phrase “implicit rank” carries a measurement connotation of the implicit association test, which has seen better days. Is this necessarily implicit? If the idea is about self-perceived rank, then that phrase could be used instead.

A: Thank you for bringing up this important point, which helped us realize that our definition of implicit rank needed clarification. In the present case, “implicit” refers to the fact that the rank is inferred from behavior and is not measured directly. “Self-perceived rank” would suggest that participants are conscious about the rank they signal through their actions, which we cannot know for certain. Thus, we opted to keep the term “implicit rank” but added a more precise definition at its first two occurrences, respectively at lines 40-41: “Our implicit rank, i.e. the rank manifested through our behavior, …” and at lines 88-90: “implicit rank (participants’ hierarchical status inferred from their behavioral choices)…”

R1-C5. Please unpack the terse verbiage in H2a and H2b; two things would help me understand this more readily: first, eliminating the dashes between words like upward-focused (which needs to be explained), and second, providing an example to make it more concrete.

A: Thank you for this suggestion. We have added more thorough explanations to hypotheses H2a and H2b and used a more illustrative language. Per Reviewer 2’s suggestions, we have also added theoretical justifications to our hypotheses and changed the term “level of social comparisons” to “level of competitive involvement,” which describes the measurement underlying these hypotheses more accurately (see our response to Reviewer 2’s Comments 3 and 4 for more detail). Hypotheses H2a and H2b have been clarified as follows: “We hypothesized that (H2a) high-ranked opponents would elicit more rivalry and admiration-seeking than low-ranked opponents, given that high-ranked opponents will be perceived as similar or superior to oneself and thus more threatening [6]. Additionally, (H2b) this effect would be further enhanced by narcissism, which is known to shift implicit rank upward [16], and (H2c) mitigated by depression, which tends to lower implicit rank [34].” (lines 97-102).

R1-C6. In the participants section, use “included” instead of “was composed of”, the latter of which involves passive voice.

A: Thank you for pointing this out. We have changed both occurrences of “was composed of” to “included” (lines 108 and 110).

R1-C7. On line 136-137, specify what type of payment. There is an imaginary currency and a real currency at play, so please be more specific in this sentence.

A: We have specified the type of payment: “… participants […] were instructed that their real money payment would be calculated based on the amount of virtual money saved on each trial …” (line 150).

R1-C8. Line 151: Spelling error on difficulty.

A: We have corrected the spelling error (line 165).

R1-C9. I was confused about rankings, specifically in line 162. Usually, being #1 is best, but here, being #200 is best, right? Also the wording is confusing to me, because “highest” implies best or most superior, but usually that is a word that belongs to the first-ranked individual. One way to simplify this and retain the numbers used in analytics, would simply be to say “best” and “worst”.

A: Thank you for this suggestion. We have corrected both occurrences of these terms (lines 176 and 253).

R1-C10. Please cite the authors of the statistics packages in R (e.g., on line 228).

A: All R packages are now duly cited (lines 233 and 243).

R1-C11. I am not sure what line 240-241 means where the authors write that “significant predictors … were maintained … even when non-significant.” Please clarify.

A: We modified the sentence as follows for clarity: “Significant predictors in Sample 1 were retained in models built for Sample 2, even when they were no longer significant in Sample 2.” (line 256).

R1-C12. The phrase “stereotypical response rates” is new to me. Is this a common phrase in the literature that I’ve missed? I am accustomed to seeing a phrase like “long-string analysis” (Curran, 2016, JESP). Either an explanation of the phrase or switching the phrase would be welcomed.

A: The term “stereotypical responding/responders” mostly occurs in the learning literature, and usually defines responses driven by the motor set and not by reinforcement (i.e. the adaptation to feedbacks received from the task). We agree that “long-string responding/responders” fits better here. We modified all occurrences of this term in the manuscript and appendix accordingly, and specified how we define “long-string” in this particular task: “… we assessed the proportion of long-string responders (defined here as participants who repeated the same choice for a given outcome measure throughout the entire task; Table C in S1 Appendix), ...” (lines 284-286).

R1_C13. On line 509, I’m not sure what the dash means after dominance.

A: The dash came with the structure of the phrase: “… self-enhancing/admiration-seeking can also be dominance-[based] not only prestige-based.” However, this sentence has been since removed in an effort to keep the discussion less philosophical and more focused on our findings (see Comment 18).

R1-C14. Figure 2 must have taken a long time to create, and it looks excellent. Nice work.

A: Thank you!

R1-C15. Figures 3 & 4 could be improved slightly by making sure the bars do not overlap. There is a setting for this in R so that you can stagger and space the bars.

A: Thank you for this thoughtful suggestion. We have edited Figures 3 and 4 accordingly.

R1-C16. There are a couple of papers that are pertinent that could be cited to round out the literature review:

a. Wallace and Baumiester had a paper on perceived opportunity for glory in narcissists.

b. Fast and Chen had a paper: https://doi.org/10.1111/j.1467-9280.2009.02452.x

A: Thank you for suggesting these references. We have included both articles in the Discussion.

a. lines 467-470: “Narcissism also correlated with faster improvement in performance on the snake arcade game (Fig G in S1 Appendix), corroborating the higher competitiveness and task engagement found in narcissism [53], especially on tasks where good performance provides the opportunity to self-enhance [54].”

b. lines 473-475: “Individuals perceiving themselves as powerful have been found to resort to aggression primarily when feeling incompetent and threatened in their self-view [55], ...”

R1-C17. In general, I would recommend pulling Chen’s papers from the literature to see if there are any other hints that would be helpful. Her work is highly relevant here.

A: Thank you for this suggestion. We have looked at Chen’s publications and found the book chapter she collaborated on (A Reciprocal Influence Model of Social Power: Emerging Principles and Lines of Inquiry) particularly relevant to our study. We now cite it in our Discussion: “Further, depressive individuals’ insecurity about their own social rank has been found to prompt competitiveness primarily out of fear of inferiority and of subsequent rejection [58], contrasting with individuals perceiving themselves as powerful, who tend to pay little attention to low-power others [59]” (lines 529-532).

R1-C18. All told, my main recommendation is for the authors to consider whether their results will hold up in the long run. This will require some reflection on the analytic path that they authors took. I think it is necessary to be explicit about whether any other analytic paths were taken, and if the number of paths is numerous, then it is probably best to hedge on the conclusiveness of this project. That being said, to the editor, I would recommend soliciting a response from the authors involving a reflection on pre-registration, an honest self-assessment of analytic paths taken, and whether the authors think

this set of results will hold.

A: Having already addressed our research project’s conceptualization and analytical approach in our response to Reviewer 1’s Comment 1, we would like to focus here on responding to how our findings may hold in the long run. As mentioned, our experiment was designed from the start to measure behavioral reactions to social defeat that we expected to be stronger in individuals high in narcissistic traits. Thus, the fact that our task's dynamics aligned with this primary hypothesis in two very different samples seems definitely encouraging about its performance in subsequent studies.

At the same time, we acknowledge that our findings' generalizability remains limited at this stage, and agree with the Editor and Reviewers about the tone of our discussion that should be more conservative and less philosophical. We have therefore removed several sections of it that may have been too far-fetched (they are marked in orange with a strikethrough in the manuscript with Track Changes). In addition, we now discuss the experiment's limitations in more detail and explain potential disparities between our behavioral measures of rivalry and admiration-seeking and Back's corresponding psychological constructs (see our answers to Reviewer 2's Comments 6 and 7 as well as lines 547-553 and lines 492-511 in the manuscript).

This research article is the first to summarize the behavioral dynamics of our experimental paradigm, and we hope that the above textual changes will encourage other studies to build on our findings, using similar or related behavioral measures. To further facilitate this, we have made the task's code freely available online for other labs (https://github.com/aszucs/cobra_task_v1; mentioned at lines 123-124 in the manuscript).

# Reviewer 2

R2-C1. My broad reaction is that the authors may be trying to do so much with this manuscript that it may be difficult for readers to extract the most meaningful information. As a result, my advice for the authors would be for them to streamline the manuscript so that it is more focused. As it currently stands, the manuscript is a bit messy and confusing because the authors have so much happening in the manuscript that it is hard to follow. For example, the authors included analyses concerning trait dominance even though the Introduction does not really provide a strong rationale for doing so since it was focused largely on narcissism and depression. Further, the authors only collected a measure of trait dominance in Sample 2 but not Sample 1 which suggests that they did not anticipate trait dominance being a central feature of this work. The authors should either drop trait dominance from their analyses (and maybe include a footnote regarding the analyses concerning trait dominance for Sample 2) or revise the Introduction so that it gives a bit more attention to trait dominance.

A: Thank you for this thoughtful and detailed comment. As acknowledged in our response to Reviewer 1’s Comment 1, we did not have strong starting hypotheses for trait dominance, and did not add this scale to the Vancouver protocol for this reason. We have now removed the effects of trait dominance from our main findings, moving them to the Exploratory analyses section, which seems more adequate. Since PLOS One does not allow footnotes, we added the table and figure illustrating findings with trait dominance to the Appendix (S1 Appendix p. 10).

We also simplified the Results’ structure in the subsection of H2 findings (lines 359-379). We hope that these changes helped to improve the Results’ overall flow and clarity.

R2-C2. I think the results concerning narcissism were the most interesting in the manuscript but it is hard to follow everything because there are so many different conceptualizations of narcissism included in the manuscript. My advice would be to simplify things. For the FFNI, it probably makes the most sense to focus on the three-dimensional model (i.e., extraversion, antagonism, and neuroticism). If the authors think it is important to also report the results for the total FFNI score and the grandiose and vulnerable dimensions, then it may be better to do that in a footnote.

A: Thank you for this suggestion. We simplified our findings by removing the grandiosity and vulnerability subscales and were happy to notice that it did not affect our conclusions as much as we thought it would, while definitely helping to make them more focused.

R2-C3. It may be helpful for the authors to provide a bit more information concerning the rationale for their hypotheses. I think the authors have very interesting ideas but it may be helpful for readers if they provide a little more information to clarify their logic for some of the predictions.

A: We have detailed the rationale behind our hypotheses, and linked them more explicitly to the concepts outlined in the Introduction:

“We expected (H1a) both rivalry and admiration-seeking to increase as a reaction to the cumulative experience of defeat throughout the task; (H1b) narcissism to further enhance both behaviors, prompting a more intense reaction to defeat [26]; and (H1c) depression to dampen them, favoring submissive responses to adversity [35].” (lines 93-97).

“We hypothesized that (H2a) high-ranked opponents would elicit more rivalry and admiration-seeking than low-ranked opponents, given that high-ranked opponents will be perceived as similar or superior to oneself and thus more threatening [6]. Additionally, (H2b) this effect would be further enhanced by narcissism, which is known to shift implicit rank upward [16], and (H2c) mitigated by depression, which tends to lower implicit rank [34].” (lines 97-102).

R2-C4. The Results section was difficult to follow. I think the authors could make it far more readable by streamlining the number of variables they are including in their analyses so I hope they consider that approach.

A: We reduced the number of variables in our main findings by removing trait dominance and simplified the Results’ structure in the subsection reporting findings related to H2 (lines 359-379). For further details, please see our response to Comment 1.

R2-C5. I was a bit confused by the operationalization of “social comparison” in the manuscript. If I am understanding it correctly, the authors used increases in point-stealing and rank-buying in conjunction with the rank of the opponent to capture “social comparison.” I think the construct that is being captured by the authors is interesting but I am not quite sure that it is really social comparison.

A: We agree with the reviewer and have changed the term “level of social comparisons” to the “level of competitive involvement”. We think that the latter is more accurate, since participants are not choosing their opponents, and the social comparisons are imposed on each trial. What depends on the participants is rather the competitive involvement these comparisons will elicit, as measured by their rivalry and admiration-seeking choices in reaction to the opponent's rank.

R2-C6. The rigged video game tournament is certainly an interesting approach for capturing these sorts of dynamics. I applaud the authors for their efforts to use this sort of approach. However, I think that some of the limitations of this approach deserve a bit more attention in the Discussion. The fact that point-stealing took place before playing an opponent whereas rank-buying took place after playing an opponent is an issue. The authors acknowledge that issue briefly but I think it deserves more attention.

Also, the issue that participants would never see the leader board again after their session makes it a bit odd and may be a somewhat weak situation with regard to motivating individuals to consider point-stealing or rank-buying.

A: Thank you for this suggestion, the task’s design indeed deserved more room in the Discussion. We now discuss in more detail potential behavioral consequences of having the point stealing decision take place before playing the opponent, and link it more clearly to the lack of association of point stealing with antagonism (please, see our answer to Comment 7 below for more detail).

We have added other potential weak points of our task’s design to the limitations: “The competition was limited to the duration of the task (participants did not have access to the league table once they finished playing) and the competition did not take place in real time (opponents were said to be past players), which may have taken away some of participants’ motivation to perform well. It is further possible that some participants did not believe in the deceptive elements incorporated in the task and therefore experienced less affective involvement in the competition.” (lines 547-553).

R2-C7. The pattern of results for FFNI antagonism were surprising. The authors briefly address this issue in the Discussion but it may warrant a bit more attention and consideration from the authors.

A: We now discuss this pattern in more detail and have put it in perspective with the self-reported construct of Rivalry: “In Back’s conceptualization, admiration-seeking roughly corresponds to agentic extraversion and rivalry to antagonism [25]. However, in our study, antagonism only enhanced rank buying, our behavioral measure of admiration-seeking (Table 2). Rivalry in our paradigm did not, in fact, include components of reactive anger, a core facet of FFNI antagonism [39], since point stealing occurred before playing against a given opponent. Further, based on the instructions, opponents were presented as previous participants and therefore were not handicapped by point stealing in real time. On the other hand, rank buying took place right after learning the trial’s outcome and thus likely acquired a reactive component. This is also corroborated by the increase of rank buying against high-ranked opponents over time (Table 1). It nevertheless remains unclear whether a similar pattern would occur with self-reported rivalry, since FFNI antagonism and rivalry measured by the Narcissistic Admiration and Rivarly Questionnaire (NARQ) are not fully overlapping constructs [25,26]. Rivalry in our paradigm matches Back’s definition, namely a willingness to surpass and devalue others in a socially insensitive way [25], and is consistently correlated with the self-reported motivation of outperforming others (Fig H in S1 Appendix). However, revenge-orientation has a strong association with rivalry in prior research [26], and it is likely that our behavioral measure of rivalry does not capture the constructs’ more antagonistic aspects.” (lines 492-509).

---

## [Editor Report · Decision Letter 1]

24 Nov 2020

Status, rivalry and admiration-seeking in narcissism and depression: a behavioral study

PONE-D-20-25628R1

Dear Dr. Dombrovski,

We’re pleased to inform you that your manuscript has been judged scientifically suitable for publication and will be formally accepted for publication once it meets all outstanding technical requirements.

Kind regards,

Peter Karl Jonason

Academic Editor

PLOS ONE
---

## [Editor Report · Acceptance letter]

26 Nov 2020

PONE-D-20-25628R1 

Status, rivalry and admiration-seeking in narcissism and depression: a behavioral study 

Dear Dr. Dombrovski:

I'm pleased to inform you that your manuscript has been deemed suitable for publication in PLOS ONE. Congratulations! Your manuscript is now with our production department. 

Kind regards, 

on behalf of

Dr. Peter Karl Jonason 

Academic Editor

PLOS ONE